# Reproductive Parameters and Host Specificity of *Eurytoma erythrinae* (Hymenoptera: Eurytomidae), a Biological Control Agent of the Erythrina Gall Wasp, *Quadrastichus erythrinae* (Hymenoptera: Eulophidae)

**DOI:** 10.3390/insects14120923

**Published:** 2023-12-03

**Authors:** Walter T. Nagamine, Juliana A. Yalemar, Mark G. Wright, Mohsen M. Ramadan

**Affiliations:** 1Hawaii Department of Agriculture, Division of Plant Industry, 1428 South King St., Honolulu, HI 96814, USA; wnaloha@yahoo.com (W.T.N.); juliana.a.yalemar@hawaii.gov (J.A.Y.); 2Department of Plant and Environmental Protection Sciences, University of Hawaii at Manoa, 3050 Maile Way, Honolulu, HI 96822, USA; markwrig@hawaii.edu

**Keywords:** biological control, *Erythrina sandwicensis*, Eulophidae, Eurytomidae, Hawaii, wiliwili

## Abstract

**Simple Summary:**

The erythrina gall wasp, *Quadrastichus erythrinae* (Hymenoptera: Eulophidae), invaded Hawaii in April 2005 and rapidly dispersed throughout all the Hawaiian Islands in less than a year. Severe infestations devastated native and introduced trees of the genus *Erythrina* (Fabaceae), destroying thousands in the Western Pacific and Hawaii. Recently, this wasp has become a serious pest of *Erythrina* species around the world, including mainland USA, Mexico, and the Neotropical region. Expensive maintenance and chemical control measures have failed to contain this gall wasp and biocontrol was thought to be the only long-term solution. Exploratory trips in Tropical East Africa, South and West Africa, and Madagascar were undertaken during December 2005–June 2006 to determine the origin of this pest and survey its associated parasitoids as potential biocontrol agents. Tanzania was selected as a focal point of exploration because of the number of *Erythrina* species listed to be endemic to that country, more than anywhere else in Africa. Three hymenopteran ectoparasitoids were found in association with the galls, a eurytomid (*Eurytoma* sp.) and two eulophids (*Aprostocetus* spp.), whose larvae develop within the galls on immature stages of the gall former. Parasitoids from the Tanzanian and South African collections were amenable for insectary rearing on EGW from Hawaii. Host specificity and biology studies were conducted on the dominant parasitoid, *Eurytoma erythrinae,* in a containment facility in Hawaii. The parasitoid was found to be specific to EGW and was approved for release with good results in reducing the impact of the pest, saving the native *Erythrina* trees in Hawaii. We report on reproductive performance, rearing biology, host specificity testing, and the implication for biocontrol elsewhere with EGW problems.

**Abstract:**

*Eurytoma erythrinae* Gates & Delvare (Hymenoptera: Eurytomidae) is an important biological control agent of the erythrina gall wasp (EGW), *Quadrastichus erythrinae* Kim (Hymenoptera: Eulophidae), an invasive species likely originating in eastern Africa that is a threat to *Erythrina* trees in Hawaii and worldwide. Thousands of *Erythrina* trees in Hawaii have succumbed to EGW since 2005 and died within a few years of infestation. The endemic wiliwili tree, *Erythrina sandwicensis*, an important component of Hawaii’s dry forests and one of few deciduous native trees, were severely impacted by this wasp. Early during the invasion by EGW it became evident that the endemic species may be driven to extinction, and exploration programs for natural enemies of the EGW started in December 2005. East Africa was selected as the starting point for natural enemy exploration owing to high native *Erythina* species richness. Several gall formers were found in Tanzania and a putative color variant type of *Q. erythrinae* was detected in association with three ectoparasitoids. During January 2006, the dominant parasitoid of this gall former was introduced to Hawaii and described as the new species, *E. erythrinae*. It was found in Ghana and South Africa attacking other gall wasp species on *Erythrina*. *Eurytoma erythrinae* was a voracious ectoparasitoid feeding as a predator on 1–5 adjacent EGW immatures to complete its development. Host specificity studies that included seven nontarget gall-forming species showed no evidence of attraction or parasitism by this parasitoid. Mean ± SEM longevity of host-deprived females (40.4 ± 2.2 days) was significantly higher than males (20.5 ± 1.1 days). Host feeding enhanced longevity of ovipositing females (51.3 ± 1.5 days). Female *E. erythrinae* is synovigenic, with high egg-maturation rate. Peak fecundity (105–239 offspring/female), host feeding biology, short life cycle (18.4 ± 0.1 days), and synchronization with the host were additional desirable attributes of this species. The parasitoid was approved for field release in Hawaii in November 2008. A total of 3998 wasps were distributed on six Hawaiian Islands, with establishment in less than a year. Impacts on high density infestations of EGW were sufficient to prevent tree deaths. Limited rates of parasitism on low-density galled leaves, flowers, and seedpods necessitated the consideration for releasing a second parasitoid, *Aprostocetus nitens* Prinsloo & Kelly (Hymenoptera: Eulophidae). We report on the reproductive characteristics and host specificity of *E. erythinae* that could be of importance for classical biocontrol programs in areas with an EGW problem.

## 1. Introduction

The erythrina gall wasp (EGW), *Quadrastichus erythrinae* Kim (Hymenoptera: Eulophidae), is an invasive species that threatens erythrina trees (Fabaceae), including the endemic wiliwili tree, *Erythrina sandwicensis* Degener, in Hawaii. This gall-forming wasp was described in 2004 as a new species from specimens collected in Mauritius, Reunion, and Singapore [1]. During April 2005, it was detected on the island of Oahu and quickly spread to all the major Hawaiian Islands by August 2005. Its worldwide distribution currently includes American Samoa, mainland China, Guam, Hong Kong, India, Okinawa, Philippines, Taiwan, and Thailand [2,3,4,5]. It was reported from the continental United States (Florida) during October 2006, subsequently Mexico, and widely throughout the Neotropical region [6,7,8]. 

*Erythrina* is a genus of 123 species of Gondwanan origin, which are distributed throughout the tropics and warm temperate regions [9,10]. Hawaii is home to the endemic wiliwili tree, *Erythrina sandwicensis*, a deciduous species that is an important component of dryland forest areas on the leeward sides of the islands. The soft, light wood of wiliwili was traditionally used for outriggers of Hawaiian canoes and for fishnet floats and surfboards. The orange and scarlet seeds are strung into valuable wreathes [9]. Wiliwili trees are also mentioned in Hawaiian legend [11]. Wiliwili is unique in that it is the only species in the genus to produce flowers of various colors (orange, red, salmon, peach, light green, yellow, or white). In response to concerns of loss of natural stands of wiliwili and possible extinction due to the EGW infestation, the University of Hawaii Lyon Arboretum’s Rare Hawaiian Plants Program established a wiliwili seed bank with representative seed collections from multiple microhabitats of the islands to preserve the species and its genetic diversity [12].

Non-native *Erythrina* species in Hawaii include *E. variegata* L., an introduced ornamental species from tropical Asia that is commonly used in landscape parks, schools, and roadways. The beauty and shade provided by these large, spreading trees have been lost as almost all these trees were killed by EGW shortly after the invasion occurred (Figure 1A–D). Only a few trees have survived, and these have been routinely treated with expensive systemic insecticides. During the onslaught of infestation, The City and County of Honolulu removed about 2000 *Erythrina* trees killed by the EGW [13]. The columnar form of *E. variegata*, known as Tropic Coral or tall *Erythrina*, was developed for use as a windbreak for soil and water conservation and for planting around farms [14]. It is highly susceptible to EGW infestations and only a few of these trees remain alive in the state (Figure 1F). Another introduced ornamental tree, *E. crista-galli* L., a South American species, showed some tolerance to the EGW and has survived despite unsightly galled flowers, stems, and seedpods [15].

The EGW is a minute gall-forming wasp belonging to the family Eulophidae (≈1.6 mm in length). Females insert their eggs (mean ± SEM = 203.6 ± 5.3 eggs/female at 30 °C, [16]) into young leaves and stems and the wasp larvae develop within plant tissue, inducing the formation of galls in leaflets and petioles (Figure 1A). As the infestation progresses, leaves curl and appear deformed, while petioles and shoots become swollen (Figure 1B). Larvae pupate within the galls (Figure 2A) and adult wasps emerge by tunneling through tissue. Heavily galled leaves and stems result in loss of growth and vigor as photosynthesis is reduced and plant health declines. Severe infestations eventually cause mortality (Figure 1D–F). 

Systemic insecticides were developed for control as the EGW larvae feed within the plant tissue. Although chemical injection systems and soil drenches were shown to be effective in some cases, they were expensive and were not cost-effective treatments for all *Erythrina* plantings or natural stands [17,18]. Conservation and chemical use were not feasible for the endemic *E. sandwicensis* in its often-remote native forest habitat [19]. The use of natural enemies was believed to be the most cost effective, sustainable, long-term solution to the EGW problem. 

A classical biological control program was initiated by the Hawaii Department of Agriculture (HDOA) during December 2005. *Erythrina* trees in Hawaii were rapidly declining from the EGW infestation so exploratory efforts were shared by the HDOA and the University of Hawaii (UH Manoa) to search the greatest area in the shortest possible time. The surveys were supported by funds from the Hawaii Department of Lands and Natural Resources. HDOA conducted explorations in Tanzania, Mozambique, and South Africa. HDOA had determined Tanzania to be the likely evolutionary origin of *Erythrina* due to the high diversity of *Erythrina* species native to the region (i.e., 15 species and subspecies, 75% of listed *Erythrina* in Tanzania) [10]. The UH made surveys in Madagascar, Mozambique, South Africa, and west African countries of Benin, Ghana, Togo, and Nigeria. All field collections of galled leaves were sent back to the HDOA Insect Containment Facility for processing.

A dominant chalcidoid natural enemy of the gall-forming *Quadrastichus* spp. wasps was found in the Morogoro region, Tanzania, in January 2006. Specimens of this wasp were sent to a specialist, Michael Gates, at the United States Department of Agriculture-Systematic Entomology Laboratory (USDA-SEL) in Beltsville, Maryland, for identification. It was determined to be a new species and was described as *Eurytoma erythrinae* [20]. This study reports on the biological attributes, host specificity, and release records of *E. erythrinae* conducted by the HDOA Plant Pest Control Branch.

## 2. Materials and Methods

### 2.1. Plant Propagation

The Indian coral tree, *Erythrina variegata*, was used for rearing EGW as it was the most abundant seed source of propagative material available and was observed to be an excellent host in Hawaii. Its seedlings were favored because of their rapid growth. It was scarified using an electric file, with nearly 100% germination within seven days, and produced usable 15–20 cm height seedlings in 3–4 weeks. Three seeds were planted in 10 cm square pots with potting mix (Miracle-Gro 6-Quart All-purpose Potting Soil Mix) and kept in outdoor screened cages (76 × 88 × 127 cm, 70 mesh) and watered daily.

### 2.2. EGW Propagation

Eight pots of *E. variegata* plants (15–20 cm height, 3 plants/pot), each placed in a 13 cm Ø saucer, were put into a screened cage (30 × 30 × 60 cm, 70 mesh). A total of about 150 pairs of EGW were placed into each cage during a three-day period. Honey (SUE BEE^®^ SPUN^®^ siouxhoney.com/sue-bee-spun-honey Sioux City, Iowa, USA) was dotted on the inside of the cage to feed the wasps. Galls began appearing on the leaves and stems of the host plants within a week and adults began emerging in about 20 days (Figure 1A and Figure 2A). Plants were watered every other day and any excess water draining into the saucer was emptied to prevent EGW adults from drowning in it.

### 2.3. Origin of the Parasitoid Colony

In December 2005–February 2006, HDOA collected gall wasps from Erythrina abyssinica Lam. ex DC. leaves in Tanzania that appeared to be a color variant of the EGW present in Hawaii. Tanzanian specimens from the Chalinze Village, Morogoro region (6°38′20.95″ S, 38°21′07.83″ E, 212 m), were sent to an eulophid expert for identification (John La Salle, CSIRO, Entomology, Canberra, Australia, identification letter to HDOA), who confirmed the specimens as Quadrastichus erythrinae. Those were later found in other villages in the Rubungo, Bwawani, Morogoro region, and Masai camp, Arusha region, Tanzania. Therefore, we reared the parasitoids from the Tanzanian collection. Infested plant leaves were shipped to Hawaii, where parasitoids emerged. Galled leaves and stem samples with evidence of parasitism were prepared for shipment to Hawaii, collected from South Africa, Kenya, Ghana, Benin, Togo, and Nigeria. Hundreds of galled leaves and stems of nine Erythrina species (*E. abyssinica* Lam., *E. caffra* Thunb., *E. crista-galli* L., *E. latissima* E. Mey., *E. lysistemon* Hutch., *E. sacluxii* Hua, *Erythrina* sp., *E. variegata* L., and *E. variegata* var. orientalis Murr.) were packed well with the cut ends of petioles wrapped in wet tissue paper and parafilm. Leaves were then folded between tissue paper to absorb the extra moisture, placed in perforated plastic bags, and, afterward, placed in a cooler with blue ice packs (Igloo Ice Blocks, Katy, TX, USA). All bags were packed in cotton pillowcases then in a cardboard box with the USDA shipping permit attached to it. Parcels were air shipped to the HDOA Insect Containment Facility in Honolulu and arrived in 2–6 days (FedEx, Tanzania, and Airway Cargo, South Africa). Experiments with the parasitoid started on the second generation in the quarantine facility (mean ± SEM, temperature 21.8 ± 0.12 °C, mean RH 70.2 ± 2.4%, light 12:12, D:L). 

### 2.4. Eurytoma Erythrinae Propagation

Galled plants were ready to be used for parasitoid propagation 14 days after exposure to the EGW females. This was conducted inside the containment facility. About 25 pairs of newly eclosed *E. erythrinae* were placed in each cage (30 × 30 × 60 cm, 70 mesh) containing eight pots of galled plants. The parasitoids were left in the cage for 14 days and then removed. A new generation of parasitoid adults began emerging in about 17 days. Daily emergence rates and sex ratios were monitored. 

### 2.5. Life History Study

Observations were made of ovipositional behavior by placing naïve mated females on a galled plant using a dissecting Nikon SMZ-745 259 stereomicroscope. Information on the duration of immature stages was obtained by periodical dissection of galls infested with immature EGW that were being preyed upon by *E. erythrinae* larvae. Mated *E. erythrinae* females were placed in a cage with pots of galled plants with parasitoids, using the same procedure as described for its propagation. After 24 h, the females were removed. Gall dissections were made to determine the developmental stadia of *E. erythrinae* immature stages. Median developing period was reported since gall dissections (about a hundred galls) were made every 2–3 days. 

### 2.6. Longevity Study and Size of Wasps

Adult longevity of host-deprived parasitoids was determined by making daily collections of newly emerged *E. erythrinae* wasps and placing them in a wide-mouth clear Mason Jar (3.8 L), with a cloth cover secured over the mouth of the jar with rubber bands. A few drops of honey were dotted inside the jar and water was provided in a glass bottle (30 mL) with a cotton wick. At least one wasp of each sex had to be present in the jar so that mating could occur. Wasps were checked daily for mortality. Front wing length, measured from tegula to the wing tip, and body length, measured from tip of the head to tip of the abdomen, were recorded for each dead adult [21]. Measurements were made for laboratory individuals and field-collected wasps were also compared with the size of EGW (Oahu Island, 10 October 2023, ex. *E. variegata*, *n* = 12). 

### 2.7. Fecundity

Fecundity tests were conducted to determine the number of progenies a female could produce in her lifetime. One female and five male *E. erythrinae* adults, all newly emerged, were placed in a large jar (cloth secured over open end) with one pot of 14-day-old, galled host plants and a few drops of honey and water was provided. Every three days, the pot of galled plants was removed from the jar and placed in a 30 cm^3^, 70 mesh, screened cage to hold for daily recording of *E. erythrinae* emergence. A new pot of galled plants was added to the jar and the procedure was repeated every three days until the *E. erythrinae* female died. Wasps were checked daily for mortality. Upon the death, the female was dissected and the number of mature eggs (Figure 2A) remaining in the ovaries were counted using a Nikon SMZ-745 259 stereomicroscope in the laboratory. Dissection for counting mature ovarian eggs of 1-day-old *E. erythrinae* females was also conducted to determine potential fecundity of newly emerged females (*n* = 5). Dissections were made in saline solution [22].

### 2.8. Host Specificity Testing

All host specificity testing for *E. erythrinae* was conducted in the HDOA Insect Containment Facility (minimum temperature 18.7 °C, maximum temperature 24.1 °C, minimum RH 61.6%, maximum RH 84.3%, light 12:12, D:L). Tests started on the second generation of parasitoid rearing. The objective was to determine if this parasitoid would attack any nontarget gall-forming insects in Hawaii. Because *E. erythrinae* attacks eulophids, we checked the Hawaiian eulophid fauna for possible nontarget host use. The endemic Hawaiian eulophid species listed in the Hawaiian Terrestrial Arthropod Checklist [23] are not gall formers and, therefore, were not included in the tests [24,25]. *Sympiesis* (*=Ophelinus*) *mauiensis* (Ashmead) is a parasitoid of Lepidoptera; *S. hawaiiensis* (Ashmead) is of unknown biology, *Pauahiana lineata* Yoshimoto and *P. maculatipennis* (Ashmead) attack lepidopterous leaf miners; *P. swezeyi* Yoshimoto is a parasitoid of *Trioza*, psyllids; and *P. metallica* Yoshimoto is of undetermined habit [26]. 

Not many gall-forming insects occur in Hawaii and could be considered for nontarget testing. The seven gall-forming insects tested included one endemic Hawaiian psyllid, four beneficial species used for weed biological control (three fruit flies of Diptera: Tephritidae and one eriococcid scale (Hemiptera: Eriococcidae)), and two immigrant wasps, one agaonid (Hymenoptera: Agaonidae) and one eulophid (Hymenoptera: Eulophidae), Table 1. Insect galls used in testing were exposed as whole plants infested in the laboratory or as fresh branch cuttings collected from the field. Galled plants were used for *Eutreta xanthochaeta* ex. *Lantana* (life cycle duration was 7–9 weeks), *Procecidochares alani* ex. Hamakua pamakani (life cycle duration 5–7 weeks), *Procecidochares utilis* ex. Maui pamakani (both life cycle duration 5–7 weeks), and *Tectococcus ovatus* ex. strawberry guava (life cycle duration 4–6 weeks). Galled branch cuttings were used for *Josephiella microcarpa* ex. Chinese banyan, *Ophelimus* sp. ex. eucalyptus, and *Trioza* sp. ex. ohia. The life cycle of gall formers was exposed as cuttings contained a range of immature stages (larva, pupae, and pharate adults), revealed by microscopic dissections of some of these galls before exposure. Attempts to infest plants in the laboratory with the latter three insect species were unsuccessful. The control insect was laboratory-reared EGW ex. *Erythrina variegata* plants. All insects tested were exposed as live immatures in their galls. Plant cuttings were placed in glass bottles (30 mL) containing a nutrient solution (Floralife, 10 g/L distilled water, www.floralife.com, accessed on 10 October 2023) to extend their freshness.

Host specificity evaluations were based on choice tests that would best represent the field situation. A plant or cutting infested with one of the seven nontarget species (test) and an EGW-infested *E. variegata* plant (control) were placed side by side in a screened cage (42 × 42 × 66 cm, 70 mesh screens). Five mated, naïve *E. erythrinae* females (≤7 d old age) were taken from a laboratory colony and placed in the cage with honey as a food source and water provided. Counts were made of *E. erythrinae* females visiting each plant and recorded separately as landing on the plant (leaves or galls). Six daily counts were made at successive hourly intervals for two consecutive days. Each count was made by observing the number of females on each plant only at that moment. The counts were started on the first day of exposure for six counts in the day (8:00 A.M.–2:00 P.M.). If any of the five parasitoids died within the first week of exposure, they were replaced with new ones. Parasitoids were then removed after two weeks, and the test and control plants were placed in separate cages to determine new parasitoid emergence. There were 4–5 replicates for each insect tested. After one month, all galls were dissected and examined under a Nikon SMZ-745 259 stereomicroscope for any evidence of parasitism by *E. erythrinae*. Dead cadavers with parasitoid oviposition or feeding marks or the presence of parasitoid immatures were revealed by dissection (Figure 2C). 

### 2.9. Colonization Records on the Islands

A table of parasitoid release on the islands with numbers of wasps released on different host plants, dates of release, and dates of establishment is provided. Parasitoids were taken from the insectary in Honolulu in screen cover vials (33 mL) with about 20 pairs per vial, placed in a cooler for field release on different islands. Infested plants and total released wasps were recorded for every island. To follow up on parasitoid recovery, infested leaves were taken to the laboratory in paper bags and cooler to Honolulu and emerged parasitoids were tallied for establishment dates. None of the Hawaiian eurytomids parasitize the EGW and differ from *E. erythrinae* in antennal and leg coloration and would thus be easily distinguished from *E. erythrinae*. Emerging *E. erythrinae* were confirmed by matching the morphological description: female color mostly black except for the yellow scape, pedicel, pro-coxae, and meso-coxae; male color black, yellow areas as described for female (Figure 2B and Figure 5C) [20]. 

### 2.10. Statistical Analysis and Vouchers

An analysis of variance was used to assess the potential significance of differences in the number of parasitoids produced by parasitism on the different hosts exposed to the wasps. Frequency of visits and parasitism were statistically analyzed using a one-way ANOVA test. Means were separated by Tukey’s standardized range honestly significant difference test and *t*-test at α = 0.05 level. Percentage data were arcsine square-root-transformed before analysis. All analyses were conducted using SAS JMP Version 11 [27].

Voucher specimens of *Eurytoma erythrinae* were deposited in the insect reference collection of the HDOA, the Bernice P. Bishop Museum, and UH insect collection, Honolulu, Hawaii. Additional depositories are USNM (National Museum of Natural History, Smithsonian Institution, Washington, DC, USA). 

## 3. Results

### 3.1. Life History

*Eurytoma erythrinae* is a biparental idiobiont solitary ectoparasitoid like many other members of Eurytomidae [28,29]. Adults of both sexes are black in color but easily distinguishable by morphology and body color from other *Eurytoma* species and those from Hawaii. The metasoma on the abdomen of the female is oval and compressed laterally, while that of the male is petiolate (Figure 2B and Figure 5C). The female inserts its ovipositor into the gall (Figure 2B) to deposit a single egg measuring 140 × 260 µm in length (Figure 2A, red arrow). The parasitoid larva feeds as an ectoparasitoid on its paralyzed host by extracting its body fluids and chewing with the mandibles, leaving the host carcasses dried, which enabled us to count the hosts killed by the parasitoid. Mandibles of the larvae are well-sclerotized with visible blades (Figure 2C). Their larvae have few body setae, but they are hairier than other chalcids. Setae are generally longest on the thoracic segments and shortest on the abdominal segments [30]. After feeding on one host, the parasitoid larva may tunnel into adjacent gall chambers and feed on other host larvae or pupae (Figure 2C). Larva may feed on the gall tissue to reach other EGW larvae. The parasitoid larva then pupates within the gall (Figure 2D). The emerged parasitic wasp tunnels its way to the outside, producing a characteristic larger exit hole than that of the EGW. The diameter of the exit hole is correlated with the width of the head capsule, the broadest body segment in this species. The maximum head width of *E. erythrinae* (male head capsule 0.55 ± 0.02 mm, and female head capsule 0.65 ± 0.017 mm) is broader than the EGW head capsule (male head capsule 0.39 ± 0.007 mm, female head capsule 0.43 ± 0.006 mm), producing significantly larger exit holes on the galls (F_3,44_ = 59.65, *p* < 0.0001). Differences are easily distinguished through a lens. 

Although *E. erythrinae* and EGW larvae are similar in color, the former has prominent sclerotized mandibles (Figure 2C) and setae on its body, while the latter does not, and their bodies have a greenish tinge of plant material visible in the gut (Figure 2A). *Eurytoma erythrinae* larval size varied according to the number of host larvae consumed. The adult parasitoids were consistently larger than the host insect (F_3,44_ = 25.77, *p* < 0.0001, Table 2), indicating their feeding on several hosts to develop to maturity (1–5 hosts). Measuring wing length as an indicator of size of wasps, the males and females of *E. erythrinae* were significantly larger than their hosts (F_3,44_ = 18.678, *p* < 0.0001, Figure 3). Furthermore, female *E. erythrinae* was significantly larger than female *A. nitens* (thelytokous eulophid) that feeds on an individual EGW host to complete development (F_4,55_ = 20.3204, *p* < 0.0001).

Based on a series of gall dissections containing parasitized EGW, the approximate duration of *E. erythrinae* life stages were as follows: egg (2–3 d), larva (11 d), and pupa (4 d); data were obtained at 21.8 ± 0.12 °C. Males and females start emerging after 16 d, with peak emergence in 19 d. The female emergence pattern declined and emergence ceased after 25 d under laboratory conditions. Male emergence followed the female’s pattern, with increasing numbers showing peaks of emergence during the bulk of female eclosion, to ensure mating success (Figure 4). 

During observations of ovipositional behavior, it was noted that the female used its mouthparts to first search the plant surface for the presence of a suitable host in a gall. The female then moved forward and used the tip of her abdomen to confirm the oviposition target area. The abdomen was then raised and the ovipositor inserted into the gall for oviposition. A small droplet of host body fluid exuded after the ovipositor was withdrawn from the gall and the female ingested it. This host-feeding behavior provides a nutritional benefit, resulting in greater female longevity than males and the host-deprived females. 

### 3.2. Longevity

There was a significant difference in longevity by gender from this test on non-ovipositing females. Females lived for 40.4 ± 2.2 days, while males lived significantly shorter, 20.5 ± 1.1 days (t_58_ = 8.12, *p* = 0.0001). Female parasitoids lived 6.8-fold longer than their EGW female hosts. Analysis of body and wing length measurements showed that females were significantly larger than males, as females probably feed on more hosts than males in the laboratory-rearing colony (t_129_ = 5.12, *p* = 0.0001). Parasitoids were significantly larger than their EGW hosts because they feed on several individuals during immature development (1–5 hosts). Longevity of the ovipositing females was (51.3 ± 1.5 days), 10 days longer than the host-deprived females. This was attributed to the host-feeding by females (Table 3). This was not observed as an effect on male longevity. 

### 3.3. Reproductive Attributes

Dissection of 1-day-old *E. erythrinae* females revealed that each female had one pair of ovaries with three ovarioles each (*n* = 5). Each ovariole contained one mature egg nearest the lateral oviducts and many immature eggs following. The ovarian egg is oval with a long “tail-like” structure at one end and a short spindle at the other end as in many *Eurytoma* species [31]. 

Females had a mean oviposition period of 37.7 ± 6.7 d and a post-oviposition period of 13.7 ± 5.4 d (Table 3). During the oviposition period, the female produced a mean of 165.3 ± 39.3 eggs. A mean of 4.3 ± 0.5 eggs was produced per day during the oviposition period. Upon death, the female had a mean of 2.3 ± 1.5 eggs remaining in its ovaries. Mean longevity of ovipositing females (51 d) was twice that of male longevity (24 d) and, notably, ovipositing females lived longer that the non-ovipositing females. This enhancement in female longevity is a result of oviposition and host feeding. The egg–adult life cycle of the progeny was 18.4 ± 0.1 days for females and 18.2 ± 0.1 d for males. The sex ratio of progeny was male-biased, with 25.9 ± 10.3% female offspring under laboratory conditions. 

### 3.4. Host Specificity Testing

Results of behavioral responses and parasitism by *E. erythrinae* in choice tests between nontarget gall formers and EGW (control) showed that this parasitoid is highly specific to EGW (Table 4). It did not emerge from any of the exposed nontargets, supporting the observations that no adults showed interest in ovipositing in them. In contrast, a range of 14–62 parasitoids eclosed from control galls. 

The number of visits to test nontarget plants by the *E. erythrinae* females were very limited, so the counts for landings on “leaves” and on “galls” were combined in data analysis. The mean number of visits per replicate was 1.0 visit for *Ophelimus* sp., 0.8 visit for *E. xanthochaeta*, 0.2 visit for *P. utilis* and *T. ovatus*, and 0 visits for *J. microcarpae*, *P. alani*, and *Trioza* sp. In comparison, mean visits to the EGW-infested control plants ranged from 6.0 to 10.6 visits per replicate (ANOVA, *p* < 0.05).

There was a significant difference for parasitism among all nontarget species compared with the control (ANOVA, *p* < 0.05). *Eurytoma erythrinae* emergence was zero for all nontarget gall formers compared to mean adult emergence of 14.8–62.2 adults per replicate for the EGW controls. Dissection of the galls of nontarget insects showed no evidence of parasitism by *E. erythrinae*, thus demonstrating that the parasitoid is extremely host-specific to the EGW.

### 3.5. Records of Field Releases

All released parasitoids were propagated in the HDOA Insectary. Table 5 has the compiled release and monitoring records on different sites on the islands of Kauai, Oahu, Maui, Hawaii, Molokai, and Lanai. Records were drafted from HDOA data for the release and establishment of *E. erythrinae*. Total parasitoids released on the islands from November 2008 to August 2010 were 3998 wasps in 84 batches of 20–150 wasps/batch. Parasitoids were established in all sites and redistributed to additional locations. On the island of Oahu, the parasitoid was released in November 2008 and was reported in February 2009 in areas where it was not released. The presence of parasitoids was observed on the same day of release on several sites, which indicates parasitoid natural dispersal from nearby release sites. The mean distance of closest release site was 6.7 ± 1.2 km, *n* = 8. 

## 4. Discussion

Here, we presented important results on reproductive biology of *E. erythrinae* and the host specificity studies that provided supporting data for safe release in Hawaii. We used a Tanzanian strain of *Eurytoma erythrinae* for this study. Variation in this wasp was found among specimens collected during the surveys from different regions in Africa. Females varied in length from 2.7 to 3.0 mm, larger than the laboratory-reared colony. In females, the fuscous area on the pro-femur can form a band in the middle half of the femur. The pro-coxa and meso-coxa may be entirely dark brown. Some South African males have a more yellowish coloration to the facial setation, and the hind-femur has a brownish band in the medial half. Male specimens from Ghana and Nelspruit, South Africa’s collection were different in their completely brown meta-femur [20]. 

*Eurytoma erythrinae* collected in Tanzania during January 2006 from hosts appeared to be a color variant of EGW in Hawaii (i.e., males were typical, female wasps exhibiting a slightly longer ovipositor, yellow genae, yellow front coxae, and four setae on cercus, MMR Unpublished). Kim et al. (2004) [1], in their description of EGW, recognized the color variation in specimens from Mauritius and Singapore. Therefore, we prioritized rearing a Tanzanian-origin colony of *E. erythrinae* in the HDOA Insect Containment Facility. 

Longevity studies showed *E. erythrinae* adults to be relatively long-lived when fed honey. Females and males survived much longer than the survivorship of their hosts. When the females had access to honey and EGW galls for host-feeding, their longevity increased to an average of 51 d. This is known in several hymenopteran where host-feeding parasitoids can replenish nutrients for enhanced fecundity and prolonged longevity during the adult stage [32]. The nutrients provided by the host body fluids very likely contributed to this extended life. This observation was further proven by longevity of males, which do not host-feed and lived a shorter period than females. 

The life cycle of *E. erythrinae* (18 d) was well synchronized with the EGW (20 d) such that parasitoids emerge when hosts are available. The 37 d oviposition period also allows the female to exert control over two generations of EGW. The ability of a larva to tunnel into adjacent gall chambers and feed on several prey is advantageous and probably contributes to the dominance of this parasitoid in the native lands (≥90% parasitism in Tanzania, MMR Unpublished). Host-feeding by *E. erythrinae* females and the resulting death of the host also add to EGW mortality. 

A male-biased sex ratio suggests that more males are needed for mating because of their shorter life span, and females may mate several times. Polyandry is common among hymenopteran parasitoids under laboratory conditions [33]. The higher percentage of males may also indicate that the female was not adequately mated and, after the sperm was used, only unfertilized (male) eggs were deposited. The sex ratio of EGW was male-biased in the field (15.6% females) and insectary colony (25.6% females) ([3,16] and unpublished field records). Adverse temperatures can decrease mating success and sperm transfer, leading to increased male sex ratios in populations of haplodiploid Hymenoptera [34]. Nonetheless, oviposition experience prior to mating may impair female’s receptivity and reduce inseminated females [35].

Host specificity studies showed *E. erythrinae* to attack only the EGW in Hawaii. Females had little attraction to plants with galls of nontarget species and clearly preferred *Erythrina* plants with EGW galls. No parasitoid emergence resulted from exposures of nontarget species to the parasitoid. All dissections indicated no evidence of parasitism. The host range in Africa may include several *Erythrina* gall formers, as revealed from our collections and the literature [36]. Therefore, host range testing in other countries may return different results from Hawaii and this should be carefully investigated when considering this parasitoid for introduction in new locations. 

Examination of host–plant relationships of EGW using 71 different *Erythrina* species in Botanical Gardens of Hawaii supported an African origin for EGW, excluding 11 countries of potential origin [15]. Based on a feeding study conducted with *Q. erythrinae* from Hawaii on the Tanzanian host plant, *Erythrina abyssinica*, Messing et al., 2009 [15] disputed that the Tanzanian collections of *Q. erythrinae* by HDOA surveys could not be resolved as the same species. Therefore, exotic natural enemies should be used cautiously [37]. The African origin remains supported only by morphological identification [5]. Lin et al., 2021 [38], in their recent study of molecular phylogeny and DNA haplotype of several EGW samples to determine the origin of EGW, suggested that the Tanzanian taxa of *Q. erythrinae* is more primitive than the other taxa in their study. They support the hypothesis that *Q. erythrinae* has an African origin and originated from Tanzania or its neighboring regions [38]. 

The parasitoid also had variant coloration. *Eurytoma erythrinae* was collected from several countries in Africa during our surveys (South Africa, Tanzania, Ghana). However, *E. erythrinae* were associated with different *Quadrastichus* species across this range, and they were probably emerging from *Q. gallicola* Prinsloo & Kelly and *Q. bardus* Prinsloo & Kelly. *Eurytoma erythrinae* was already known in the South African National Collection of Insects and was considered an inquiline of unknown habit [36]. 

The success of the released parasitoid was exceptional in Hawaii and considered by some biocontrol researchers as the best in decades, with the added benefit that it improved the public perception of classical biocontrol in Hawaii [39,40]. Results of several years of post-monitoring showed the parasitoid is well established and naturally dispersing everywhere on the islands, reducing the impact of EGW on the native wiliwili (Figure 5A–F, photos before and after release) [39]. *Eurytoma erythrinae* saved *E. sandwicensis* trees from destruction and possible extinction, without expensive toxic chemical control [41,42]. The successful biocontrol of EGW in Hawaii offers opportunities for other countries to import this parasitoid. Japan was the first to introduce *Eurytoma erythrinae* from Hawaii for a biocontrol program on the islands of Sumuzy, Isigaci, and Irimuti, where the flower of *E. variegata* is the symbol of the Okinawa Region [37]. The mainland USA has two native *Erythrina* species attacked by EGW (*E. herbacea* L. and *E. flabelliformis* Kearney) in need of biocontrol introductions. The neotropics are a center of endemism for *Erythrina* and 24 species are native to Mexico. They are susceptible to EGW and should be considered at risk [43]. 

**Figure 5 insects-14-00923-f005:**
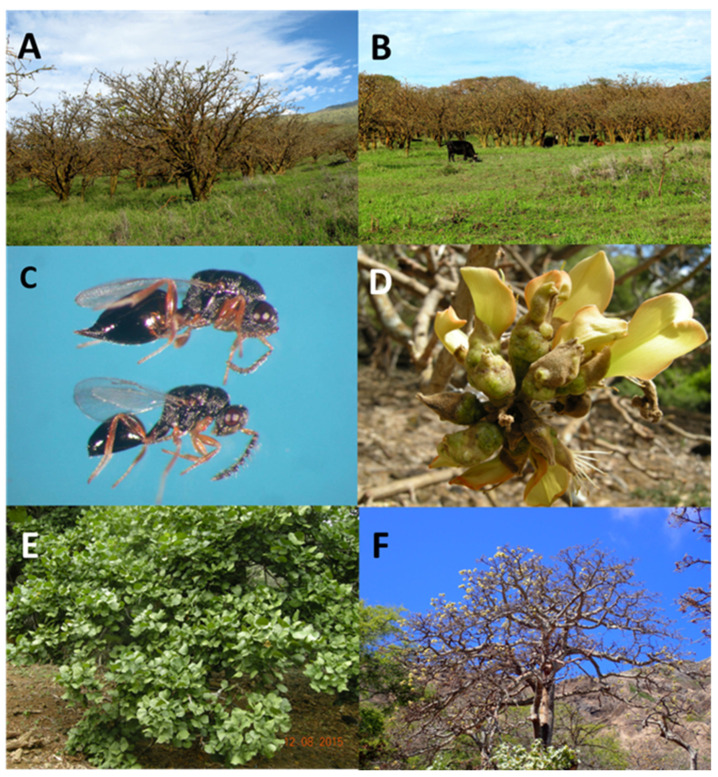
(**A**) Dried wiliwili forest *Erythrina sandwicensis* O.Deg, Lahaina Lua, Maui Island, December 2008; (**B**) massive infestation on Ulupalakua Maui Island, December 2008; (**C**) lateral habitus of *Eurytoma erythrinae* released, female (upper), male (lower); (**D**) infestation on flowers and seedpods of wiliwili still needs higher parasitism, reason for release of a second parasitoid; (**E**) wiliwili recovery after wasp establishment at Koko Head crater, Oahu Island, 8 December 2015; (**F**) wiliwii yellow flowering tree at Koko Head crater, Oahu Island, December 2020 photo. Photos credited to Mach Fukada, Maui Island Entomologist (**A**,**B**), Walter Nagamine (**C**), and Juliana Yalemar (**D**–**F**).

Kaufman, et al., 2020 [41], in their monitoring study of *E. erythinae* in Hawaii, highlighted the limited rates of parasitism of galls on flowers and seedpods (Figure 5D) and supported the release of a second parasitoid, *Aprostocetus nitens*. An application with the HDOA for final permits allowing the release of *A. nitens* from the Containment Facility is in progress at the time of writing. Studies have shown no adverse effects of *A. nitens* on any nontarget species in Hawaii. 

## 5. Conclusions

*Eurytoma erythrinae* is an ectoparasitoid that acts as a predator feeding on several immature hosts within the same gall structure during its lifetime. Adults host-feed and live much longer than host-deprived females, which enhances their rapid natural establishment in all infested areas of EGW. On the island of Oahu, the parasitoid was released and reported established within three months. Their preference for aggregated galls owing to their predatory habit and utilization of multiple individuals as prey is also a limitation of this parasitoid with low gall density, such as occurs on flowers and seed sprouts of *Erythrina* spp. This constraint supports the selection of the second parasitoid *A. nitens* to be tested and released in conjunction with *E. erythrinae*. *Aprostocetus nitens* is always smaller in size, indicating its likely development on a single host (=individual galls) such as those on the flowers and seedpods. Hawaii is planning the release of this second parasitoid with permits pending. A third parasitoid native to Africa, *Aprostocetus exertus* La Salle, has a long ovipositor sheath, longer than the body length of the female, that may reach deeply protected hosts and should possibly also be considered for release. It may be effective in parasitizing immature stages of the gall wasps residing deep inside the stems that cannot be reached by parasitoids with shorter ovipositors [44].

## Figures and Tables

**Figure 1 insects-14-00923-f001:**
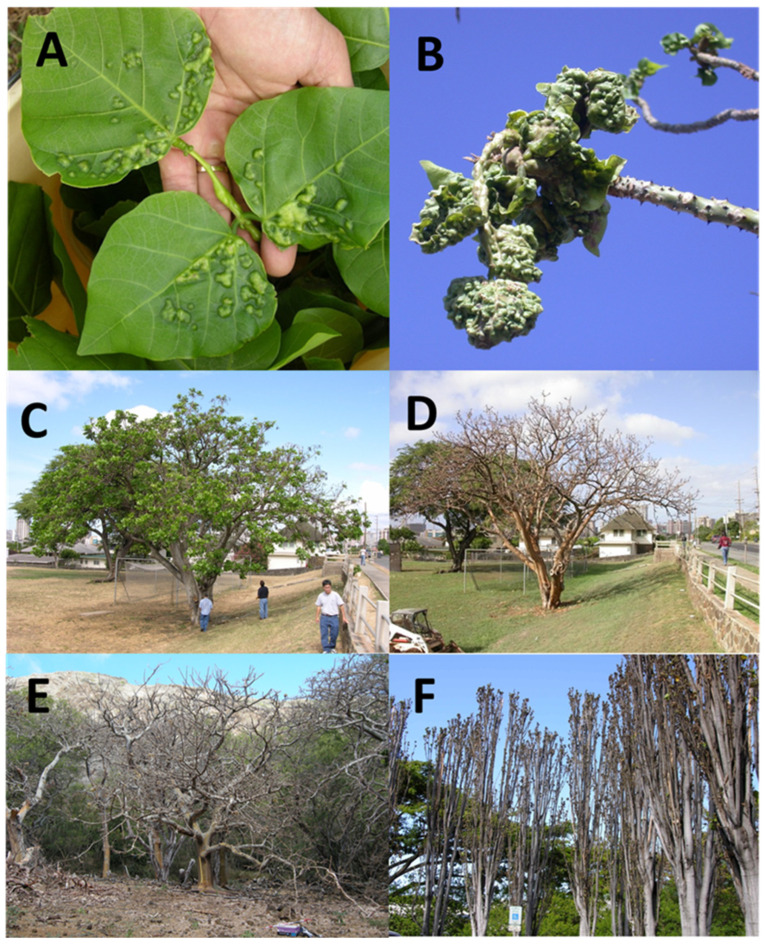
(**A**) EGW early infestation of *Erythrina variegata*; (**B**) massive infestation on *E. variegata*; (**C**) *Erythrina variegata* tree at original site of EGW detection in Manoa, Oahu Island, April 2005; (**D**) same *E. variegata* tree in Manoa, dead after heavy EGW infestation, December 2006; (**E**) dead native infested trees of *Erythrina sandwicensis*, Koko Head, Oahu; (**F**) dead infested trees of tall variety *Erythrina variegata*, Oahu Island.

**Figure 2 insects-14-00923-f002:**
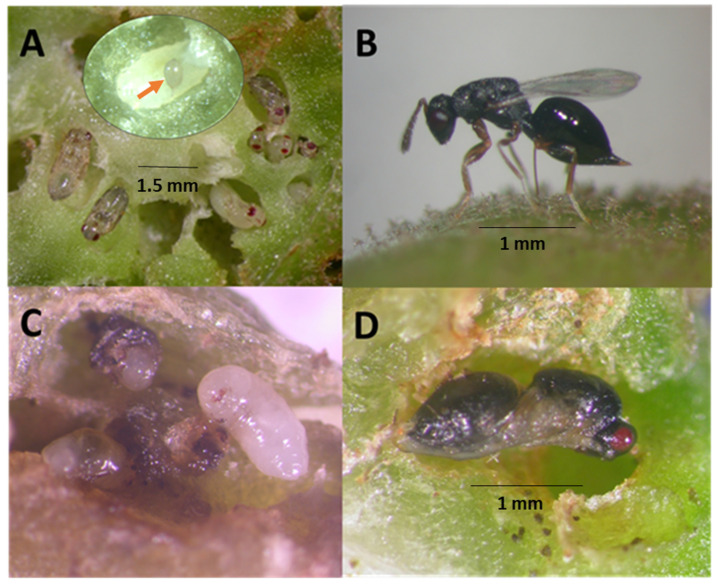
(**A**) EGW dissected from infested swollen plant tissue with numerous EGW pupae in closeness of gall chambers. Inset: an egg (red arrow) of the parasitoid on a parasitized larva; (**B**) female *Eurytoma erythrinae* ovipositin on a gall; (**C**) larva of the parasitoid devouring EGW immatures in the tunnels; (**D**) pupa of the parasitoid in a chewed chamber.

**Figure 3 insects-14-00923-f003:**
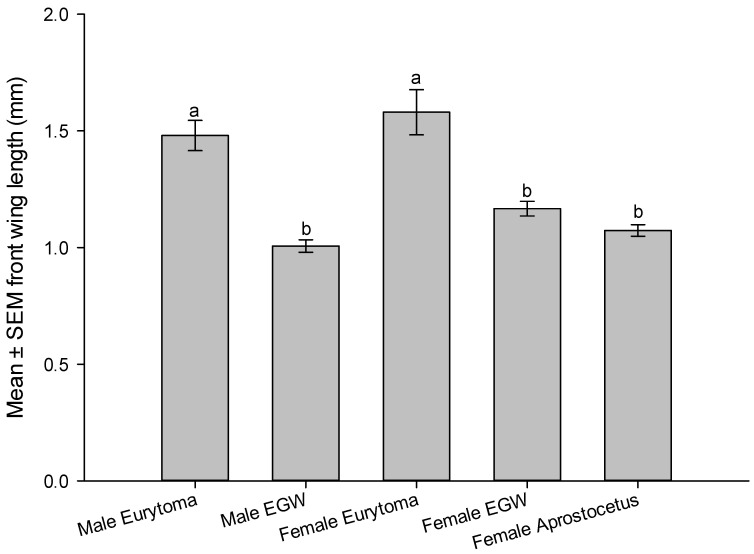
Comparison of front wing lengths (measured from tigula to tip of wing) of *Eurytoma erythrinae*, *Aprostocetus nitens*, and their host EGW, as indication of wasp’s size. Except for *A. nitens* (thelytokous laboratory colony), species and sex are wild type collected from *Erythrina variegata* on Oahu Island (12 October 2023). Bars topped by the same letter indicate no significant differences among wing lengths at the 5% level according to Tukey test.

**Figure 4 insects-14-00923-f004:**
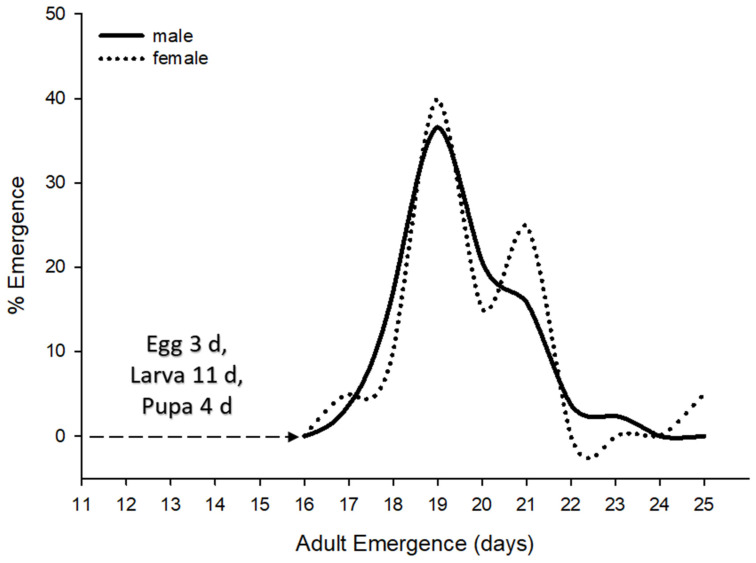
Average immature developmental period (days) and male–female emergence patterns of *Eurytoma erythrinae* under laboratory conditions; data were obtained at 21.8 ± 0.12 °C.

**Table 1 insects-14-00923-t001:** Gall-forming insects used in host specificity tests for *Eurytoma erythrinae*.

Order and Family	Gall-Former (Scientific and Common Name)	Gall-Former Status and Source	Host Plant (Scientific, Common Name), Infested Plant Part Used for Testing
Hymenoptera: Agaonidae	*Josephiella microcarpae* Beardsley & Rasplus, Banyan gall wasp	Immigrant, Field-collected, Honolulu, Oahu	*Ficus microcarpa* Chinese banyan, cuttings
Hemiptera: Eriococcidae	*Tectococcus ovatus* Hempel biocontrol agent	Biocontrol agent, Lab-reared, USFS, HVNP and HDOA	*Psidium cattleianum* strawberry guava, whole plants
Hymenoptera: Eulophidae	*Ophelimus* sp. Eucalyptus gall wasp	Immigrant, Field-collected, Camp Maluhia, Maui Island	*Eucalyptus* sp. *Eucalyptus,* Cuttings
Hemiptera: Psyllidae	*Trioza* sp. Ohia psyllid	Endemic, Field-collected, Aiea and Manoa, Oahu Island	*Metrosideros polymorpha* Ohia, Cuttings
Diptera: Tephritidae	*Eutreta xanthochaeta* Aldrich Lantana gall fly	Biocontrol agent, Field-collected, Hauula, Oahu Island and lab-reared HDOA	*Lantana camara* Lantana, whole plants
Diptera: Tephritidae	*Procecidochares alani* Steyskal Hamakua pamakani gall fly	Biocontrol agent, Field-collected, Nuuanu, Oahu Island and lab-reared HDOA	*Ageratina riparia* Hamakua pamakani, whole plants
Diptera: Tephritidae	*Procecidochares utilis* Stone Maui pamakani gall fly	Biocontrol agent, Lab-reared, UH-Manoa and lab-reared HDOA	*Ageratina adenophora* Maui pamakani, whole plants

**Table 2 insects-14-00923-t002:** Mean longevity of non-ovipositing *Eurytoma erythrinae* adults that were fed honey and their adult size by gender in comparison to the host EGW reared on *Erythrina variegata*.

Parasitoid and Host	N	Longevity (Days) Mean ± SEM	Body Length ^a^ (mm) Mean ± SEM	
			Laboratory Colony	Wild Wasps ^b^
Female *Eurytoma*	39	40.4 ± 2.2	2.1 ± 0.05	2.2 ± 0.08 a
Male *Eurytoma*	92	20.5 ± 1.1	1.5 ± 0.04	2.04 ± 0.07 a
Mean comparison		T_58_ = 8.12, *p* = 0.0001	T_128_ = 9.20, *p* = 0.0001	
Female EGW	100	5.9 ± 0.3	-	1.6 ± 0.03 b
Male EGW	100	7.8 ± 0.3	-	1.45 ± 0.07 b
Mean comparisons		T_197_ = 4.106, *p* = 0.0001	-	F_3,44_ = 25.55, *p* < 0.0001

^a^ Measured from tip of head to tip of abdomen. Body length may not accurately represent adult size due to shriveling of the specimen after death. ^b^ Wild wasps (10 October 2023, Oahu wasps ex. *E. variegata*). Means followed by a different letter in the wild wasp’s column are significantly different (ANOVA, *p* < 0.05).

**Table 3 insects-14-00923-t003:** Reproductive attributes for mated *Eurytoma erythrinae* females and their progeny. The data are shown for three replicates in which a female was fed honey and allowed to oviposit in EGW-infested plants that were replaced every three days.

Reproductive Parameter	Mean ± SEM	Range	Unit
Oviposition period ^a^	37.7 ± 6.7	31–51	days
Post-oviposition period	13.7 ± 5.4	3–20	days
Mature ovarian eggs at death	2.3 ± 1.5	0–5	number eggs
Female longevity	51.3 ± 1.5	49–54	days
Male longevity (*n* = 15) ^b^	24.6 ± 2.7	3–42	days
Female progeny	40.0 ± 15.3	20–70	adult
Male progeny	125.3 ± 41.8	82–209	adult
Total progeny per female	165.3 ± 39.3	105–239	offspring
Daily progeny	4.3 ± 0.5	3.4–4.9	adult
Sex ratio (% females)	25.9 ± 10.3	12.6–46.1	% female
Sex ratio (% males)	74.1 ± 10.3	53.9–87.4	% male
Female lifecycle (*n* = 120)	18.4 ± 0.1	15–24	days
Male lifecycle (*n* = 376)	18.2 ± 0.1	15–26	days

^a^ The day of first oviposition could not be determined because plants were exposed for 3-day intervals. Oviposition period was therefore calculated from the day of female emergence. In all three replicates, progeny emerged from the first exposure of galled plants. ^b^ For each replicate, five males were included with the female to assure mating.

**Table 4 insects-14-00923-t004:** Behavioral responses and parasitism by *Eurytoma erythrinae* females in choice tests between nontarget gall formers (test) and *Quadrastichus erythrinae* (control). Counts of *E. erythrinae* visits to galled plants were a combined number of landings on leaves and on galls.

Galled Plant Infested with Insect	Frequency of Visits by *E. erythrinae* (Mean ± SEM)	*E. erythrinae* Adult Emergence (Mean ± SEM)
*Josephiella microcarpae*	0 b	0 b
*Q. erythrinae* (control)	7.6 ± 2.6 a	25.0 ± 9.6 a
*Tectococcus ovatus*	0.2 ± 0.2 b	0 b
*Q. erythrinae* (control)	10.6 ± 3.4 a	62.2 ± 17.9 a
*Ophelimus* sp.	1.0 ± 1.0 b	0 b
*Q. erythrinae* (control)	7.2 ± 1.4 a	19.8 ± 3.1 a
*Trioza* sp.	0 b	0 b
*Q. erythrinae* (control)	7.6 ± 1.2 a	25.4 ± 9.8 a
*Eutreta xanthochaeta*	0.8 ± 0.3 b	0 b
*Q. erythrinae* (control)	6.0 ± 0.7 a	14.8 ± 5.8 a
*Procecidochares alani*	0 b	0 b
*Q. erythrinae* (control)	7.0 ± 1.8 a	14.5 ± 5.4 a
*Procecidochares utilis*	0.2 ± 0.2 b	0 b
*Q. erythrinae* (control)	9.8 ± 3.2 a	17.0 ± 6.0 a

Means followed by the same letter for each pair of tested plants separately are not significantly different (*p* > 0.05).

**Table 5 insects-14-00923-t005:** Colonization records of *Eurytoma erythrinae* on infested *Erythrina* species and establishment records by island.

Island	*Erythrina* Species Infested by the Gall Wasp	N ^2^	Numbers of *Eurytoma* Released	Release Time (Month/Day/Year)	Dates of Recovery (Month/Day/Year)
**Oahu**	*Erythrina sandwicensis* O.Deg.		1070	25 November 2008–23 August 2010	6 February 2009–23 August 2010 ^3^
	*Erythrina crista-galli* L.		470		
	*Erythrina variegata* L. ^1^		130		
Sub-total and % of total		38	1670 (41.8%)		
**Maui**	*Erythrina sandwicensis*		380	17 December 2008–23 December 2009	25 February 2009–21 January 2010
	*Erythrina crista-galli*		60		
Sub-total % of total		10	440 (11.0%)		
**Hawaii**	*Erythrina sandwicensis*.		688	1 December 2008–27 January 2010	14 January 2009–27 January 2010 ^3^
	*Erythrina crista-galli*		30		
	*Erythrina variegata* ^1^		330		
Sub-total and % of total		20	1048 (26.2%)		
**Kauai**	*Erythrina sandwicensis*		240	4 December 2008	3 March 2009–30 June 2010
	*Erythrina variegata*		30		
Sub-total and % of total		5	270 (6.7%)		
**Molokai**	*Erythrina sandwicensis*		390	14 April 2009–9 June 2009	10 February 2010–24 March 2010
	*Erythrina crista-galli*		60		
	*Erythrina variegata* ^1^		30		
Sub-total and % of total		9	480 (12.0%)		
**Lanai** total and % of total	*Erythrina sandwicensis*	2	90 (2.2%)	12 May 2009	17 February 2010–7 April 2010
Total released on all islands		84	3998		

^1^ *Erythrina variegata* L. and its variety, tall windbreak wiliwili, http://www.hear.org/issues/wiliwilionmaui/ accessed on 23 March 2022. ^2^ N = number of lots, range (20–150 wasps/lot). ^3^ Field establishment day same as release day when parasitoids found already on site before release.

## Data Availability

The data presented in this study are available on request from the corresponding author.

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
