# Peer review of "Reproductive Parameters and Host Specificity of Eurytoma erythrinae (Hymenoptera: Eurytomidae), a Biological Control Agent of the Erythrina Gall Wasp, Quadrastichus erythrinae (Hymenoptera: Eulophidae)"

_insects, 2023, doi:10.3390/insects14120923_

Round 1

Reviewer 1 Report

Comments and Suggestions for Authors

The Eurytoma is a synovigenic species, it would be good to use that term early on in the description of its biology.  Synovigeny is linked to host feeding, oosorption and dispersal capabilities. 

A male biased sex ratio in the lab may also be affected by temperature.  You might mention that in the discussion along with the other two salient explanations about a higher ratio of males. 

Overall, this is a good paper.  The demography methods and results are well done.  There is very good biological information and very good information on the biological control process.

Comments on the Quality of English Language

There are several grammatical errors and wording issues throughout.  Without line numbers it is difficult to go through and point them out.  One example is the second sentence of the simple summary.  The writing could be tightened up a little with editing: Severe infestations devastated native and introduced trees of the genus Erythrina (Fabaceae) destroying thousands of trees in the Western Pacific and Hawaii.

Author Response

Thank you for reviewing our manuscript. 

Reviewer 2 Report

Comments and Suggestions for Authors

The study by Nagamine et al provides interesting information about the classical biological control of the EWG in Hawaii . The manuscript is well organised and the presentation of the methods and results is sound and clear for the reader. I believe that it is a very nice contribution in the field of biological control. I only have some comments below for the authors to consider to clarify and improve the manuscript.

Section 2.5 life history study, how many females per cage. How many plants in the cage. How many gall dissection per dissection day

Line 398, data about exit holes?

Line405, the size its not a prove of that.

Lines 409-411, why not provide more solid data with mean and SDs?

Lines 576-577. No strong evidence for this . could you please elaborate on that? Besides the size difference were there any other evidence for that? I don’t see how you compare the size of the two insects as an indication since one is a herbivore and the other is a carnivore. Huge difference.

Lines 583-585. This sentence actually is a contradiction of the previous one. Is it a haplodiploid species?

Lines 598-602. This sentence is unclear. Please rephrase.

Line 612, do you mean E. erythrinae parasitoids? Not clear

Line615, the same as above

Author Response

Thank you so much for reviewing our manuscript.

Reviewer 3 Report

Comments and Suggestions for Authors

This manuscript reports various biological parameters (rate of development, longevity, fecundity, host specificity etc.) and the results of field release of Eurytoma erythrinae, an insect parasitoid of the invasive erythrina gall wasp, Quadrastichus erythrinae. The experiments and observations were well planned and conducted; the data were correctly analyzed and clearly presented. The reported data can be important for biological control of the gall wasp and therefore the paper can be published. I have only some minor comments and corrections.

Line 2: delete comma after “parameters”

Lines 31-32: Replace “Host specificity studies and biology” by “Host specificity and biology studies”

Line 52: delete comma after “Ghana”

Line 60: delete comma after “host”

Line 75: Replace “was determined” by “were determined”.

Lines 133-134: replace “@” by some preposition and close the parenthesis.

Lines 409-412 (text) and 448 (figure legend): please, remind here to the reader that these data were obtained at 21.8 ± 0.12 C (line 228).

Table 2: “Female Eurytoma   39” should not be in bold font.

Table 3: “Sex ratio (% males)” is not needed; the percentage of females is enough.

Line 515: These letters indicate the difference between all data in the column or for each pair of tested plants separately?

Author Response

(The authors gave the same response as above.)
